Dieting alleviates hyperuricemia and organ injuries in uricase-deficient rats via down-regulating cell cycle pathway

Yu Yun 1
Wan Xulian 2
Li Dan 1
Qi Yalin 3
Li Ning 3
Luo Guangyun 3
Yin Hua 3
Wang Lei 1
Qin Wan 2
Li Yongkun 2
Li Lvyu 4
Duan Weigang deardwg@126.com 3
1 School of Basic Medicine, Kunming Medical University , Kunming , Yunnan , China
2 School of Chinese Medicine, Yunnan University of Traditional Chinese Medicne , Kunming , Yunnan , China
3 School of Basic Medicine, Yunnan University of Traditional Chinese Medicine , Kunming , Yunnan , China
4 The Third Affiliated Hospital, Yunnan University of Traditional Chinese Medicine , Kunming , Yunnan , China
Gillespie Joseph
Electronic publication date: 2023 Sep 8
Publication date: 2023
Volume: 11
Electronic Location ID: e15999
Received 2023 Apr 19; Accepted 2023 Aug 9
Copyright: ©2023 Yu et al.
Copyright year: 2023
Copyright holder: Yu et al.
License: This is an open access article distributed under the terms of the Creative Commons Attribution License, which permits unrestricted use, distribution, reproduction and adaptation in any medium and for any purpose provided that it is properly attributed. For attribution, the original author(s), title, publication source (PeerJ) and either DOI or URL of the article must be cited.
License URL: https://creativecommons.org/licenses/by/4.0/

Keywords: Uricase-deficient rats, Hyperuricemia, Serum uric acid, Dieting, Fasting, Renal injury, Cell cycle pathway

Funding: National Natural Science Foundation of China 82260886 Yunnan Provincial Science and Technology Department–Applied Basic Research Joint Special Funds of Kunming Medical University 202101AY070001-007 Yunnan Provincial Science and Technology Department–Applied Basic Research Joint Special Funds of Yunnan University of Traditional Chinese Medicine 202101AZ070001-010 202101AZ070001-093 202101AZ070001-242 The work was financially supported by the National Natural Science Foundation of China (No. 82260886), the Yunnan Provincial Science and Technology Department–Applied Basic Research Joint Special Funds of Kunming Medical University (202101AY070001-007), and the Yunnan Provincial Science and Technology Department–Applied Basic Research Joint Special Funds of Yunnan University of Traditional Chinese Medicine (202101AZ070001-010, 202101AZ070001-093, 202101AZ070001-242). The funders had no role in study design, data collection and analysis, decision to publish, or preparation of the manuscript.

==============================
Dieting is a basic treatment for lowering hyperuricemia. Here, we aimed to determine the optimal amount of dietary food that lowers serum uric acid (SUA) without modifying the dietary ingredients in rats. Increased SUA was found in food-deprived 45-day-old uricase-deficient rats (Kunming-DY rats), and the optimal amount of dietary food (75% dietary intake) to lower SUA was established by controlling the amount of food given daily from 25% to 100% for 2 weeks. In addition to lowering SUA by approximately 22.5 ± 20.5%, the optimal amount of dietary food given for 2 weeks inhibited urine uric acid excretion, lowered the uric acid content in multiple organs, improved renal function, lowered serum triglyceride, alleviated organ injuries (e.g., liver, kidney and intestinal tract) at the histological level, and down-regulated the Kyoto Encyclopedia of Genes and Genome (KEGG) pathway of the cell cycle (ko04110). Taken together, these results demonstrate that 75% dietary food effectively lowers the SUA level without modifying dietary ingredients and alleviates the injuries resulting from uricase deficiency or hyperuricemia, the mechanism of which is associated with the down-regulation of the cell cycle pathway.

Introduction

Gout is a common disease in modern society (Yang, Zhang & Zhou, 2022). Although the underlying mechanism of gout is not yet fully understood, hyperuricemia is widely accepted to be the most important initiating factor (FitzGerald et al., 2020). Hyperuricemia is a condition characterized by an elevated serum uric acid (SUA) level. The normal upper limit of SUA is 68 µg/mL, while a level of >70 µg/mL is considered saturated and may be accompanied by symptoms (George & Minter, 2022).

Uric acid is an end product of purine metabolism in humans, and uricase (urate oxidase, Uox) deficiency almost inhibits the conversion of uric acid to allantoin (Jiang et al., 2021), a more soluble substance. Uric acid is directly transformed from xanthine by xanthine dehydrogenase (Xdh) and is excreted through urine and feces (Gao et al., 2022b). It is estimated that more than 80% of uric acid is synthesized from endogenous purines while less than 20% is obtained from the diet (Basseville & Bates, 2011; Liu et al., 2019). Additionally, more than 80% of uric acid is excreted through urine, and less than 20% is excreted through feces (Basseville & Bates, 2011; Gao et al., 2022b). Elevated SUA levels can occur due to its increased synthesis or decreased excretion. Drugs that inhibit Xdh or increase uric acid excretion can be used to lower SUA in clinical practice.

However, drugs used to lower SUA, such as Xdh inhibitors (allopurinol Whelan et al., 1979 and febuxostat Becker et al., 2005) and uricosuric agents (probenecid Yonetani, Ishii & Iwaki, 1980 and benzbromarone Shinosaki et al., 1992), are highly dependent on absorption, which can lead to severe adverse reactions (Hoyer et al., 2021; Jmeian et al., 2016; Lee et al., 2008). According to the guidelines recommended by the American College of Rheumatology (ACR) in 2020 (FitzGerald et al., 2020), basic treatments, such as adjusting diet structure and restricting body weight, are important to lower SUA levels. In addition to avoiding high-purine diets, the amount of dietary food can be used to restrict body weight and SUA. Fasting (the deprivation of food) is a common strategy (Hooshiar, Yazdani & Jafarnejad, 2023) to control body weight, although prolonged fasting is unhealthy. Unfortunately, it has been reported (Gao et al., 2022a; Miyake et al., 2014) that with the loss of body weight, fasting increases rather than decreases SUA levels, and how to adjust the optimal amount of dietary food is a challenge that remains to be solved.

The results of our previous preliminary study, which were presented in a local journal (Qi et al., 2021), demonstrated that 75% dietary food can lower the SUA level in rats. Based on these findings, we sought to repeat the study to establish the protective effects of this diet on the digestion system, intending to unveil the underlying mechanism in uricase-deficient (Uox −/−) rats (Kunming-DY [KDY] rats).

Materials and Methods

As this study is follows on from our previous research, the materials used, and the procedures of animal housing, sample collection, biochemical assays, the abundance of genes expressed in organs, and the Kyoto Encyclopedia of Genes and Genomes (KEGG) pathway enrichment analysis are similar to those reported previously (Gao et al., 2022b).

Materials

Wild-type 45-day-old Sprague-Dawley (SD) rats were obtained from Chengdu Dossy Experimental Animals Co., Ltd, Chengdu, China (Certification No. SCXK [Chuan] 2008–24). KDY rats (45 days old) were provided by our laboratory (Yu et al., 2020). Rat food was made from corn, soybean, wheat, and fish meal, containing crude protein (≥20%), fat (≥4%), crude fiber (≤5%), water (≤10%), ash (≤8%), calcium (1.0–1.8%), phosphorus (0.6–1.2%), lysine (≥1.32%), and methionine and cystine (≥0.78%). The food was produced by Suzhou Shuangshi Experimental Animal Food Co., Ltd., conforming to the Chinese standard for experimental rat food (GB-14924.3-2010).

The materials used was similar to those reported in our previous study (Gao et al., 2022b). Uric acid assay kits using the phosphotungstic acid method (Lot: C012-1-1), creatinine (Cr) assay kits using the sarcosine oxidase method (C011-2-1), urea assay kits using the urease method (C013-2-1), glucose assay kits using the glucose oxidase method (A154-1-1), protein assay kits of bicinchoninic acid (BCA) method (A045-3-2), high-density lipoprotein (HDL) assay kits (A112-1-1), low-density lipoprotein (LDL) assay kits (A113-1-1), total cholesterol (TC) assay kits (A111-1-1), and triglyceride (TG) assay kits (A110-1-1) were all purchased from Nanjing Jiancheng Bioengineering Institute (Nanjing, China). A TRIzol Plus RNA Purification kit was purchased from Invitrogen (Carlsbad, CA, USA). Hematoxylin-eosin (HE) staining kits, mouse anti- β-actin antibody (BM0627), rabbit anti-CASP3 antibody (BA2142), rabbit anti-CDK1 antibody (PB9533), rabbit anit-Ki67 antibody (M00254-4), and horseradish peroxidase (HRP) conjugated goat anti-rabbit/mouse IgG (H+L) (BA1056) were all purchased from Boster Biological Engineering Co., Ltd (Wuhan, China). Tablets provided in glass vials containing complete protease inhibitors (11873580001) were purchased from Merck & Co., LTD (Chengdu, China).

Ultrapure water was produced with a Milli Q water purification system manufactured by EMD Millipore Group (Darmstadt, Germany); the NanoDrop ND-1000 spectrophotometer used for experiments was manufactured by PeqLab (Erlangen, Germany). The multiple microplate reader (K6600A) was manufactured by Beijing Kai’ao Technology Development Co., Ltd. (Beijing, China). The gel imager (JP-K600 Plus) was manufactured by Shanghai Jiapeng Technology Co., Ltd. (Shanghai, China). All other instruments or reagents used in the study were made in China, unless otherwise stated.

Study design

Considering that diet is an important contributor to SUA, we systematically screened out how the optimal amount of dietary food affects SUA. First, KDY rats were fasted to observe whether their SUA levels increased. If their SUA levels increased, another experiment was designed to establish the optimal amount of dietary food that could lower SUA levels. Subsequently, further experiments were designed to investigate how dieting lowers the SUA level. Additionally, the histological changes in the liver, kidney and intestinal tract induced by the optimal amount of dietary food were observed (Fig. 1). The designers and animal experimenters were aware of the group allocation, while the other members of the team were only aware of the group allocation at the end of the experiment. The experimenters who assayed the indices were unaware of the group allocation before the experiment was completed.

Figure 1 Study design.

OADF, the optimal amount of dietary food; ADF, the amount of dietary food; SUA, serum uric acid; KDY rats, uricase-deficient rats on the background of Sprague-Dawley rats.

Animal housing

KDY rats were provided by our laboratory (Yu et al., 2020). The animal housing protocol was similar to that outlined in our previous study (Gao et al., 2022b). Briefly, adult male KDY rats were mated with the female for 3 weeks to generate offspring. The offspring were breastfed to the age of 3 weeks by their mothers before the mothers and male and female offspring were separated into three cages. The male offspring were included in the study when they were 45 days old, at which point they weighed approximately 180–220 g.

The rats were maintained under 22 ± 1 °C at a humidity of 45–55% under natural light and were provided free access to water and scheduled food. All animal experiments were approved by the Animal Care and Use Committee of Kunming Medical University (Approval No. KMMU-2021578) and were conducted under the Guidelines for the Ethical Review of Laboratory Animal Welfare of China (GB/T35, 892–2018). Animals were treated following the ARRIVE guidelines (https://arriveguidelines.org).

At the end of the experiments, all of the living rats used in groups were allowed to resume a free diet and water for 1 day. On the second day, they were gently treated to calm down before being intraperitoneally anesthetized with chloral hydrate (1.0 g/kg). During a state of deep anesthesia, the animals were euthanized by dislocating their necks. The rat bodies were collected in yellow plastic bags and kept in a refrigerator at −20 °C until removal by a green company for cremation.

Fasting and dieting experiments

In fasting experiments, twelve 45-day-old male KDY rats were equally randomized to control and fasting groups using random numbers. Rats in the control group were treated with free access to food and water, while those in the fasting group were provided no food but free water for 2 days. Blood (approximately 200 µL) was drawn daily to prepare serum samples. The rat’s body weight and the volume of water consumed daily were recorded, and the excreted urine was collected daily using metabolic cages. The uric acid in serum and urine was assayed using the uric acid assay kits.

In dieting experiments, animals were scheduled to receive different amounts of dietary food, namely, 25%, 50%, 60%, 70%, 75%, 80%, 90%, and 100% dieting groups. Each group was randomly arranged with six male 45-day-old KDY rats. Animals in the 100% dieting group were given free access to food and water for 2 weeks, and the amount of food consumed daily were recorded. The amount of dietary food consumed by other groups daily was discounted at 25%, 50%, 60%, 70%, 75%, 80%, and 90%. The body weight was recorded daily, and blood (approximately 200 µL) was drawn every week to prepare serum samples. The feces and urine excreted in the 24 h before treatment and on the 7th and 14th days were collected using metabolic cages. The uric acid in serum, urine and feces was assayed using the uric acid assay kits.

Sample collection

Serum sample collection

Similar to the results of our previous study (Gao et al., 2022b), serum samples were collected from the rats on scheduled days. To obtain blood samples, the rats were kept in small cages, and blood samples (approximately 0.2 mL) were drawn by cutting their tail tip (approximately two mm) without anesthesia at a local temperature of 28–32 °C. As soon as the blood had coagulated, serum was collected by centrifugation at 3,000 g for 5 min at 4 °C.

Urine and fecal sample collection

Similar to our previous study (Gao et al., 2022b), rats were kept individually in metabolic cages to collect their 24-h excreta. Urine and fecal samples were collected in a cold insulation box on ice. The urine was stirred to a homogeneous state, and 1.0 mL was sampled as soon as possible. The original urine sample was quickly diluted 20 times with sodium bicarbonate solution (50 mmol/L) to obtain the final sample for the uric acid assay. The fecal samples were weighed and mixed with three times their weight of 100 mmol/L Tris solution, before stirring the mixture on a shaker at 100 rpm for 4 h to extract the uric acid in the feces. Finally, the mixture was centrifuged at 5,000 g for 5 min at 4 °C, and the supernatant was collected for the uric acid assay.

Organ sample collection

Similar to our previous study (Gao et al., 2022b), at the end of the animal experiment, the scheduled animals were sacrificed, and their abdominopelvic and thoracic cavities were opened. Subsequently, their blood, organs and tissues, including the kidney, liver, pancreas, stomach, spleen, jejunum, adrenal gland, ileum, lung, colon, thymus, heart, and skeletal muscle (gluteus maximus) were harvested. Animal organs were weighed, cut into pieces, and homogenized with 10 × sodium bicarbonate solution (50 mmol/L) on ice. The homogenate was centrifuged at 10,000 g for 10 min at 4 °C to obtain the supernatant.

The supernatant collected from the homogenized fresh tissues and the samples for the uric acid assay were kept at −20 °C or assayed as soon as possible to prevent the false elevation of uric acid levels by Xdh (Watanabe et al., 2014).

Biochemical assays

Similar to our previous study (Gao et al., 2022b), the uric acid in the abovementioned samples was assayed using the uric acid assay kits. The assay kits had a good quantification range of uric acid from 3.91 µg/mL to 125 µg/mL. If the uric acid in the sample exceeded this range, the sample was diluted and assayed again (Gao et al., 2022b; Qi et al., 2021). The protocols of the uric acid assay kits are available at http://www.njjcbio.com/uploadfile/product/big/20220824104754751.pdf.

The levels of Cr, urea, glucose, HDL, LDL, TC, and TG in the serum were determined using the Cr, urea, glucose, HDL, LDL, TC, and TG assay kits, respectively. The protocols can be downloaded from http://www.njjcbio.com/uploadfile/product/big/20220824104736657.pdf (for the Cr assay kits), http://www.njjcbio.com/uploadfile/product/big/20220824104836688.pdf (for the urea assay kits), http://www.njjcbio.com/uploadfile/product/big/20220824095620250.pdf (for glucose assay kits), http://www.njjcbio.com/uploadfile/product/big/20220824090614994.pdf (for HDL assay kits), http://www.njjcbio.com/uploadfile/product/big/20220824090652930.pdf (for LDL assay kits), http://www.njjcbio.com/uploadfile/product/big/20220824090536011.pdf (for TC assay kits), and http://www.njjcbio.com/products.asp?id=2579 (for TG assay kits).

Abundance of genes expressed in organs and kegg pathway enrichment

The protocol is similar to that reported in our previous study (Gao et al., 2022b). Briefly, at the end of the experiment, the animals (59 days old) were anesthetized with urethane (1.0 g/kg), and their liver and kidney were harvested. Subsequently, the fresh organs were frozen with liquid nitrogen and ground into powders. The total RNA in the powders was extracted and purified using the TRIzol Plus RNA Purification kit. The quantity and quality of RNA were measured using the NanoDrop ND-1000 spectrophotometer, and the RNA integrity was assessed using denaturing gel electrophoresis of RNA as previously described (Chen et al., 2017; Yin et al., 2015).

Double-stranded cDNA (ds-cDNA) was reverse-transcribed from the total RNA using a SuperScript ds-cDNA synthesis kit (Invitrogen, Carlsbad, CA, USA) in the presence of 100 pmol/L oligo dT primers. The Solexa high-throughput sequencing technique was used to sequence the cDNA by Sangon Biotech Co. Ltd. (Shanghai, China). The raw data containing reads of 150-base nucleotides in the fastq format were transformed to the original sequences in fasta format using Seqkit software in a disk operation system (DOS) model (Shen et al., 2016). The sequences matching ≥27 bp of the rat’s reference mRNA sequences ( https://www.ncbi.nlm.nih.gov/) were screened out using TBtools software (v0.664445552). The expected value of fragments per kilobase of transcript sequence per million base pairs sequenced (FPKM) was used to normalize the expression level (Lin et al., 2017; Trapnell et al., 2010).

The expression abundance of a gene in the organs between groups was compared using Student’s t-test. The significant genes (P<0.05), either up- or down-regulated, were enriched to determine KEGG pathways (https://www.kegg.jp/) of interest. KEGG pathway enrichment analysis (Kanehisa & Goto, 2000) was performed using the ClusterProfiler software (3.0.5 version) (Yu et al., 2012). The P- and Q-values were also calculated using the software.

Western blot

The Western blot protocol was similar to that reported in our previous study (Yu et al., 2020). Briefly, rat tissue samples (approximately 100 mg) were homogenated in icy isotonic lysis buffer (25 mmol/L Tris, pH 7.4, 150 mmol/L NaCl, complete protease inhibitors from Roche). The protein supernatant was obtained by centrifuging the homogenate at 5,000 g for 5 min at 4 °C. The protein concentration of the supernatant was assayed using the protein assay kit of the BAC method and adjusted to the same concentration. The protein sample was mixed with gel loading buffer, boiled for 5 min for denaturation, and applied to 10% sodium dodecyl sulfate-polyacrylamide gel electrophoresis (SDS-PAGE). The protein (20.0 µg) was separated by a direct current at 100 V. The separated protein in the SDS-PAGE was transferred to a nitrocellulose (NC) membrane by a direct current at 10 V for 60 min in a semi-dry electrophoretic transfer cell. The NC membrane was blocked with 3% bovine serum albumin (BSA) at an ambient temperature for 2 h before bathing in the primary antibody solution (1:400–1,000) at 4 °C for 3 h. The membrane was rinsed with TST Buffer (20 mmol/L Tris–HCl, pH 7.5, and 0.05% Tween-20) for 10 min, before bathing in a solution of secondary antibody (1:1,000) for another 2 h. The NC membrane was rinsed three times with TST buffer for 10 min and analyzed using the ECL detection kits. The band signals were recorded by the gel imager. The band brightness was quantified with ImageJ software (1.48 V). The relative brightness was calculated using Equation (1). (1) Relativebrightness=BrightnessbandofinterestedproteinBrightnessbandofβ−actin

Histological examination

At the end of the experiment, the rats were anesthetized with chloral hydrate (1.0 g/kg). Subsequently, their abdominopelvic and thoracic cavities were opened, and a normal saline solution for injection was perfused into their left ventricle immediately at 150 cmH2O. The perfusate was discharged via the right atrium by making a small hole with small scissors. After the perfusion of normal saline solution (approximately 200 mL), a 4% neutral methanal solution (approximately 200 mL) was perfused at 150 cmH2O. Then, organs, including the liver, kidney, duodenum, terminal ileum and the start of the colon, were harvested. The organs were fixed in a 4% methanal solution for more than 24 h, before subjecting to HE staining. The organs were immersed in a 4% methanal solution until routine HE staining was conducted. The organs were adjusted and embedded in paraffin after dehydration with anhydrous ethanol and xylene. Sections of the paraffin-embedded organs were cut at a 5-µm thickness and stained using HE staining kits. Finally, the stained sections were visualized and scanned using a fluorescence microscope in a light mode.

Statistical analysis

All data obtained were included in the study. The animals used in the experiment were the same age, but had various weights. To account for the difference caused by body weight, the indices referring to total quantity were corrected by their body weight. Additionally, to balance individual differences between animals, the values before treatment (Day 0 or W 0) were defined as “1” in the long-term observance experiments. The values are expressed as the mean ± standard deviation (STDEV.S for a sample from a population, or STDEV.P for a whole population). If the values were found to have a normal distribution by the normality test (Shapiro–Wilk test), Student’s t-test was performed to compare the means between groups using the T-TEST function of Excel software (19.0); otherwise, a chi-test or rank sum test was applied using SPSS for Windows 16.0. Statistical significance was accepted at P < 0.05.

Results

Elevation of SUA levels by fasting

The SUA level of KDY rats in the control group was relatively stable (Fig. 2A). Fasting for 1 day or more significantly increased the SUA level (Fig. 2A). Surprisingly, the urine uric acid concentration in the fasting group was significantly decreased (Fig. 2B), as was the total uric acid in 24-h urine (Fig. 2C). The fasted animals drank less water (Fig. 2D), excreted less urine (Fig. 2E), and eventually showed a significant reduction in body weight (Fig. 2F). The results were similar to those of our previous study (Qi et al., 2021).

Figure 2 Fasting experiment (mean ± STDEV.S, n = 6).

In male KDY rats (uricase-deficient rats), fasting increased serum uric acid (SUA) level (A), decreased uric acid concentration in urine (UUA) (B), and decreased the total uric acid in 24-h urine (24-h UUA) (C). Additionally, fasted animals drank less water (24-h water intake) (D) and excreted less urine (24-h urine) (E), and eventually significantly reduced body weight (F). Control, control group, KDY rats were given free access to food and water; Fasting, fasting group, rats were deprived of food and only given free access to water. *P < 0.05 vs control group at the same day, Student’s t-test, two-tailed.

SUA levels were affected by different amounts of dietary food

In rats with 100% dieting (free access to water and food), the amount of consumed food increased (Fig. 3A) with increasing days, and based on the standard, the amount of dietary food was discounted on the same day-age in other groups. Different amounts of dietary food had different effects on SUA (Fig. 3B). A 2-week duration of 75% dieting significantly decreased the SUA levels in the rats, while the amounts of dietary food more or less than 75% did not significantly reduce SUA levels, and in some cases, even tended to increase SUA levels to certain extent. If the SUA levels before the experiment were set as “1” the effect of 75% dieting on lowering SUA levels was manifested (Fig. 3C).

Figure 3 Different amounts of dietary food affected serum uric acid (SUA) levels and body weight in male KDY rats (mean ± STDEV.S, n = 6).

(A) In the group of 100% dieting (free access to food and water), the amount of consumed food increased with increasing days. (B) A 2-week of 75% dieting significantly decreased SUA levels, while the amounts of dietary food more or less than 75% did not significantly reduce SUA levels, occasionally, even tended to increase SUA levels to certain extent. (C) If the SUA levels before experiment were set as “1”, and the effect of 75% dieting in lowering SUA levels was manifested. D, Dietary food less than 75% resulted in body weight loss while those more than 75% facilitated body weight increasing. *P < 0.05 vs W0, Student’s t-test, two-tailed. W0, just before treatment; W1, treated for one week; W2, treated for two weeks; KDY rats, uricase-deficient rats on the background of Sprague–Dawley rats.

In addition, reductions in dietary food consumption did not facilitate increases in body weight and even led to the loss of body weight in some cases. However, 75% dieting resulted in the maintenance of a stable body weight (Fig. 3D). The results were similar to those of our previous study (Qi et al., 2021).

SUA levels were lowered by 75% dietary food

Another six male KDY rats were used to carefully evaluate how 75% dieting lowers the SUA level. After 1-week treatment of 75% dieting, the SUA levels were significantly lowered, which continued into week 2 (Fig. 4A). However, the uric acid concentration in urine significantly decreased after treatment for ≥1 week (Fig. 4B), while that in feces was almost unchanged (Fig. 4C). Additionally, the total uric acid excreted through urine significantly decreased (Fig. 4D), while that excreted through feces varied without significance (Fig. 4E). The total uric acid, excreted through both urine and feces, significantly decreased (Fig. 4F).

Figure 4 Male KDY rats (uricase-deficient rats) affected by 75% dieting (mean + STDEV.S, n = 6).

After 1-week treatment of 75% dieting, the SUA levels were significantly lowered, which continued into week 2 (A). However, the uric acid concentration in urine (UUA) significantly decreased after treatment for ≥ 1 week (B), while the uric acid concentration in feces (FUA) was almost unchanged (C). The total uric acid in 24-h urine (24-h UUA) significantly decreased (D), and that excreted through feces (24-h FUA) varied without significance (E). The total uric acid excreted through urine and feces (24-h TUA) significantly decreased (F). Their body weight after 1 week of treatment slightly decreased and slightly increased after 2 weeks of treatment (G). The reduction of food supply significantly reduced the volume of consumed water (H), which resulted in decreased 24-h urine (I). However, the 75% dieting tended to reduce the amount of feces excreted (24-h feces), although without significance. *P < 0.05 vs W0, Student’s t-test, two-tailed. W0, just before treatment; W1, treated for 1 week; W2, treated for 2 weeks.

In parallel, the body weights of rats significantly decreased after 1 week of treatment and slightly increased after 2 weeks of treatment (Fig. 4G). Reductions in food supply significantly reduced the volume of water consumed by the rats (Fig. 4H), which resulted in decreased urine volume (Fig. 4I). However, 75% dieting tended to reduce the amount of feces excreted, although without significance (Fig. 4J).

Uric acid content in organs was lowered by 75% dietary food

We next determined the uric acid levels in the organs of KDY rats in the control and 75% dieting groups, including the kidney, liver, pancreas, stomach, spleen, jejunum, adrenal gland, ileum, lung, colon, thymus, heart, skeleton muscle, and brain (Fig. 5). The organs were listed based on the organ uric acid content from the highest to lowest in the control group. Dieting with a 75% food reduction over 2 weeks significantly reduced the uric acid content in most organs. The kidney and liver in the control group were the two organs containing the highest levels of uric acid. However, the uric acid levels in these two organs were significantly lowered by a 2-week 75% dietary food treatment. Surprisingly, the uric acid content in the adrenal glands tended to increase, although without significance.

Figure 5 Uric acid content in the organs of KDY rats (uricase-deficient rats) treated by 75% dieting for two weeks (mean + STDEV.S; n = 6).

Control, control group, KDY rats were given free approach to food and water; 75% dieting, 75% dieting group, rats were given 75% food of control group in the same day and given free access to water. *P < 0.05 vs control group, Student’s t-test, two-tailed.

Renal functions and glycolipid metabolism were improved by 75% dietary food

Serum Cr and serum urea nitrogen (BUN) are the two indices that tend to be used to evaluate renal function. In the present study, the 2-week 75% dietary food treatment significantly lowered the level of serum Cr (Fig. 6A) but had almost no effect on BUN (Fig. 6B). Moreover, 75% dietary food for 1 week lowered the serum glucose level, while the dietary food for 2 weeks had almost no effect on serum glucose (Fig. 6C) and serum HLD (Fig. 6D). The dietary food tended to increase serum LDL (Fig. 6E) and decrease serum TC (Fig. 6F) and TG (Fig. 6G).

Figure 6 Serum biochemical indices affected by 75% dieting (mean + STDEV.S, n = 6).

Two weeks of 75% dieting significantly lowered the level of serum creatinine (Cr) (A), but almost had no effects on BUN (serum urea) (B). One week of 75% dieting lowered serum glucose level, but two weeks of 75% dieting almost had no effect (C). The dieting for two weeks had no effect on HLD (D). The dieting tended to increase LDL (E), and decrease TC (F) and TG (G). W0, just before treatment; W1, treated for 1 week; W2, treated for 2 weeks. HDL, high-density lipoprotein; LDL, low-density protein; TC, total cholesterol; TG, triglyceride. *P < 0.05 vs W0, Student’s t-test, two-tailed.

Gene products associated with urate metabolism were affected by the 2-week 75% dietary food treatment

As the liver is one of the most important organs to generate uric acid, and the kidney is the main organ for uric acid excretion, the gene expression in these two organs at the mRNA level was assayed after 2-week treatment using a high throughput sequencing technique. Based on the current understanding (Xu et al., 2017; Yun et al., 2017), the gene products associated with uric acid transportation, synthesis, degradation, and purine recycling were determined (Table 1) and their expression abundance was calculated.

Table 1 Gene products associated with urate metabolism affected by 75% dieting for 2 weeks (FPKM, mean ± STDEV.S, n = 3).

FPKM, fragments per kilobase of transcript sequence per million base pairs sequenced as exons. Control, male KDY rats had free approach to food and water. A total of 75%, rats were given 75% dietary food for 2 weeks.

Gene	Liver	Kidney	Note	
	Control	75% dieting	P	control	75% dieting	P		
Abcc4	2.49 ± 0.24	1.16 ± 0.18	↓0.002	12.89 ± 10.56	21.35 ± 1.77	0.243	Uric acid secretion	
Abcg2	3.75 ± 0.90	8.52 ± 0.41	↑0.001	39.49 ± 16.68	47.76 ± 7.97	0.482	
Lgals9	119.16 ± 18.41	125.83 ± 5.74	0.581	18.18 ± 10.78	13.89 ± 3.47	0.547	
Slc17a1	8.95 ± 0.96	26.67 ± 1.42	↑0.000	29.69 ± 17.77	29.39 ± 4.97	0.979	
Slc22a6	0.31 ± 0.21	0.06 ± 0.06	0.110	202.63 ± 136.62	448.57 ± 68.08	↑0.049	
Slc22a12	0.04 ± 0.02	0.01 ± 0.02	0.122	143.18 ± 51.42	203.53 ± 63.05	0.268	Uric acid intake or reclamation	
Slc22a13	0.01 ± 0.02	0.02 ± 0.01	0.796	10.01 ± 3.12	8.62 ± 4.48	0.683	
Slc22a8	93.71 ± 14.38	89.31 ± 0.96	0.626	161.08 ± 115.93	430.3 ± 99.53	↑0.038	
Slc2a6	0.61 ± 0.33	0.57 ± 0.13	0.869	3.05 ± 0.51	2.48 ± 0.26	0.157	
Slc2a9	10.86 ± 0.09	6.35 ± 0.23	↓0.000	3.40 ± 2.70	5.13 ± 0.45	0.335	
Ada	1.81 ± 0.31	1.87 ± 0.17	0.790	14.22 ± 2.14	10.92 ± 0.35	0.058	Uric acid synthesis	
Xdh	61.43 ± 4.72	22.86 ± 3.50	↓0.000	19.67 ± 14.81	42.69 ± 4.2	0.061	
Gda	10.21 ± 1.71	6.43 ± 2.08	0.072	4.66 ± 3.57	6.56 ± 0.60	0.416	
Aprt	72.28 ± 5.22	27.86 ± 1.51	↓0.000	274.45 ± 220.98	146.32 ± 6.15	0.372	Purine recycling	
Hprt1	13.84 ± 2.05	10.72 ± 0.51	0.063	12.87 ± 6.17	11.38 ± 1.27	0.703	
Pnp	220.69 ± 15.23	129.77 ± 3.07	↓0.001	96.25 ± 35.53	122.38 ± 11.53	0.292	
Aox1	20.94 ± 2.80	105.77 ± 2.31	↑0.000	2.09 ± 2.04	0.86 ± 0.59	0.374	Uric acid degradation	
Uoxa	205.56 ± 22.91	367.8 ± 18.34	↑0.001	0 ± 0	0.01 ± 0.02	0.374	
Notes.

↓, down-regulation

↑, up-regulation

Bold, P < 0.05 vs. control, student’s t-test, two-tailed.

a Expression of Uox in KDY rats was an invalid transcript, as exons 2–4 were deleted.

The 2-week 75% dietary food treatment significantly changed the abundance of many gene products in the liver. Indeed, Abcc4 (related to urate secretion), Slc2a9 (related to urate intake), Xdh (related to urate synthesis), Aprt (related to purine recycling), and Pnp (related to purine recycling) were significantly down-regulated. Additionally, Abcg2 (related to urate secretion), Slc17a1 (related to urate secretion), Aox1 (related to urate degradation), and Uox were up-regulated (Table 1). The Uox gene expressed in KDY rats was an invalid transcript of uricase due to the deletion of exons 2–4 (Yu et al., 2020). In addition to Uox, Aox1 (Xu et al., 2020) (aldehyde oxidase 1) is regarded as another enzyme that catalyzes uric acid, although with lower efficiency, and the Aox1 gene was found to be up-regulated in the livers of KDY rat in the 75% dieting group.

However, fewer gene products were affected in the kidneys of KDY rats in the 75% dieting group (Table 1), and only Slc22a6 (a transporter for urate secretion) and Slc22a8 (a transporter for urate reclamation) were up-regulated.

Differential KEGG pathways were affected by 75% dietary food

A total of 32,623 genes were quantitatively sequenced in the liver at the mRNA level. As a result, we identified 9,520 gene products that were differentially expressed in the liver of the 75% dieting group versus those of the control group. According to the pathways recorded in the KEGG library, 40 down-regulated pathways and 29 up-regulated pathways were enriched in the livers of the 75% dieting rats with both P- and Q-values <0.05. The top 10 down- and up-regulated pathways are listed in Table 2. Although the 2-week 75% dietary food treatment induced down- or up-regulation of many pathways, no differential pathways were directly associated with urate metabolism. The most characteristic functional changes in the liver were associated with the down-regulation of the cell cycle (ko04110) or apoptosis, and the regulation of material and energy metabolism.

Table 2 Differential KEGG pathways enriched in the livers in the 75% dieting group versus those of the control group with both P- and Q-value below 0.05 (n = 3).

No.	id	Description	Significant	Annotated	P-value	Q-value	Regulation	
1	ko04110	Cell cycle	46/1021	120/7817	1.96E−12	3.92E−10	down	
2	ko04111	Cell cycle - yeast	32/1021	73/7817	7.80E−11	7.80E−09	down	
3	ko04115	p53 signaling pathway	29/1021	68/7817	1.34E−09	8.93E−08	down	
4	ko03013	RNA transport	47/1021	177/7817	8.90E−07	4.45E−05	down	
5	ko04120	Ubiquitin mediated proteolysis	40/1021	145/7817	2.06E−06	8.23E−05	down	
6	ko03460	Fanconi anemia pathway	20/1021	52/7817	3.50E−06	0.000117	down	
7	ko04068	FoxO signaling pathway	35/1021	128/7817	1.10E−05	0.000314	down	
8	ko04113	Meiosis - yeast	20/1021	60/7817	4.15E−05	0.000921	down	
9	ko04214	Apoptosis - fly	20/1021	60/7817	4.15E−05	0.000921	down	
10	ko04921	Oxytocin signaling pathway	39/1021	158/7817	4.65E−05	0.000929	down	
1	ko01230	Biosynthesis of amino acids	22/570	90/7817	3.09E−07	6.09E−05	up	
2	ko00270	Cysteine and methionine metabolism	13/570	45/7817	1.19E−05	0.000725	up	
3	ko00140	Steroid hormone biosynthesis	16/570	66/7817	1.43E−05	0.000725	up	
4	ko03320	PPAR signaling pathway	18/570	81/7817	1.55E−05	0.000725	up	
5	ko01200	Carbon metabolism	24/570	131/7817	2.11E−05	0.000725	up	
6	ko00680	Methane metabolism	10/570	29/7817	2.21E−05	0.000725	up	
7	ko00100	Steroid biosynthesis	8/570	19/7817	2.80E−05	0.000786	up	
8	ko03010	Ribosome	46/570	346/7817	4.26E−05	0.000859	up	
9	ko04146	Peroxisome	18/570	87/7817	4.31E−05	0.000859	up	
10	ko00220	Arginine biosynthesis	8/570	20/7817	4.37E−05	0.000859	up	

A total of 32,623 genes at the mRNA level were quantitatively sequenced in the kidney, and 1,689 gene products were found to be differentially expressed in the kidneys of the 75% dieting rats. A total of eight down-regulated pathways and nine up-regulated pathways in the 75% dieting rats were enriched with both P- and Q-values <0.05 (Table 3). Surprisingly, almost no differential pathways were directly associated with urate synthesis and transportation. The most characteristic functional changes were also associated with the down-regulation of the cell cycle (ko04110) and the regulation of material and energy metabolism, similar to the kidney to some extent.

Table 3 Differential KEGG pathways enriched in the kidney of the 75% dieting group versus those of the control group with both P- and Q-value below 0.05 (n = 3).

No.	id	Description	Significant	Annotated	P-value	Q-value	Regulation	
1	ko04110	Cell cycle	16/134	120/7817	1.54E−10	1.90E−08	down	
2	ko03460	Fanconi anemia pathway	10/134	52/7817	1.34E−08	8.26E−07	down	
3	ko04111	Cell cycle - yeast	9/134	73/7817	3.75E−06	0.000154	down	
4	ko04115	p53 signaling pathway	8/134	68/7817	1.88E−05	0.00058	down	
5	ko04113	Meiosis - yeast	7/134	60/7817	6.74E−05	0.00166	down	
6	ko04114	Oocyte meiosis	8/134	112/7817	0.000644	0.013227	down	
7	ko04974	Protein digestion and absorption	7/134	94/7817	0.001102	0.019393	down	
8	ko03440	Homologous recombination	4/134	29/7817	0.001406	0.021645	down	
1	ko03320	PPAR signaling pathway	7/103	81/7817	8.79E−05	0.010637	up	
2	ko04915	Estrogen signaling pathway	7/103	100/7817	0.00033	0.018349	up	
3	ko04010	MAPK signaling pathway	11/103	252/7817	0.000455	0.018349	up	
4	ko00982	Drug metabolism - cytochrome P450	5/103	55/7817	0.000743	0.021212	up	
5	ko00980	Metabolism of xenobiotics by cytochrome P450	5/103	57/7817	0.000876	0.021212	up	
6	ko04152	AMPK signaling pathway	7/103	126/7817	0.001317	0.026562	up	
7	ko00830	Retinol metabolism	5/103	68/7817	0.001946	0.032784	up	
8	ko00260	Glycine, serine and threonine metabolism	4/103	42/7817	0.002167	0.032784	up	
9	ko04920	Adipocytokine signaling pathway	5/103	74/7817	0.002826	0.03801	up	

To verify the down-regulation of the cell cycle pathway, some gene products were evaluated at both the mRNA and protein levels. Among them, caspase 3 (Casp3), cyclin-dependent kinase 1 (Cdk1) and Ki67, which positively participate in the cell cycle pathway, were significantly down-regulated at both levels (Fig. 7).

Figure 7 Down-regulation of gene products associated with the cell cycle pathway in 75% dieting KDY rats’ liver for 2 weeks (mean + STDEV.S; n = 3).

(A) Casp3, Cdk1 and Ki67 in the liver was evaluated at mRNA level, and a high value of FPKM means a high expression; (B) Casp3, Cdk1 and Ki67 in the liver was evaluated by Western blot; C, Results of B were quantified by relative brightness. FPKM, Fragments Per Kilobase of exon model per Million mapped fragments. Control, control group, KDY rats were given free approach to food and water; 75% dieting, 75% dieting group, rats were given 75% food of control group at the same age in days and given free access to water. *P < 0.05 vs control group, Student’s t-test, two-tailed.

Organ injuries were alleviated by the 2-week 75% dietary food treatment at the histological level

Uricase deficiency caused multiple mild organ injuries in KDY rats (Fan et al., 2021), similar to the phenomena observed in uricase-deficient mice (Guo et al., 2019). In the control group, swollen hepatic cells were observed, and the sinuses were crowded with swollen hepatic cells, which caused the hepatic sinuses to be very narrow (Fig. 8A). In the 75% dieting group, the swollen hepatic cells were recovered to some extent, and the hepatic sinuses were manifested (Fig. 8B).

Figure 8 The injured liver in KDY rats was alleviated by 75% dieting for 2 weeks.

(A) The liver in the control group; (B) the liver in the 75% dieting group. Black arrow showed the typical vacuole around the hepatic nucleus (glycogen depositions), and the red arrow showed the sinus. Bar = 80 µm.

The edge of the kidney in the control group was round, and its cortex was swollen (Fig. 9A), while that in the 75% dieting group was more like an ellipse, with some extent of cortex recovered (Fig. 9B). The glomeruli in the control group were enlarged, the edges of the glomeruli were unclear, the renal capsules were filled with red material, and proliferative stromal cells were frequently observed (Fig. 10A). The glomeruli in the 75% fasting group were restored to some extent, the edges of the glomeruli were clear, almost nothing was observed in the renal capsules, and fewer proliferative stromal cells were observed (Fig. 10B). In the renal medulla, the tubule lumens in the control group were filled with red material (Fig. 11A), while those in the 75% dieting group were much clearer (Fig. 11B).

Figure 9 The injured kidney in KDY rats was alleviated by 75% dieting for 2 weeks (overview).

(A). The kidney in the control group; (B). The kidney in the 75% dieting group.

Figure 10 The injured kidney in KDY rats was alleviated by 75% dieting for 2 weeks (glomeruli in the renal cortex).

(A) The kidney in the control group; (B), the kidney in the 75% dieting group. Red arrow showed the edge of typical glomerulus and its capsule; blue arrow showed the typical proliferative stromal cells. Bar = 80 µm.

Figure 11 The injured kidney in KDY rats was alleviated by 75% dieting for 2 weeks (tubules in the renal medulla).

(A) The kidney in the control group; (B) the kidney in the 75% dieting group. A blue arrow showed a typical lumen filled with red material. Bar = 80 µm.

The intestinal tracts in the control group, including the duodenum, ileum, and colon, were relaxed, as evidenced by their baggy outlines observed in transverse view (Figs. 12A–12C), and the intestinal villi were short or even lost (Figs. 12H–12H). On the contrary, the 2-week 75% dietary food treatment restored the intestinal tension (Figs. 12D–12G) and lengthened the intestinal villi (Figs. 12K–12M).

Figure 12 The injured intestinal tract was alleviated by 75% dieting for 2 weeks.

(A & H) The duodenum in the control group; (B & I) the ileum in the control group; (C & J) the colon in the control group; (D & K) the duodenum in the 75% dieting group; (E & L) the ileum in the 75% dieting group; (G & M) the colon in the 75% dieting group.

Discussion

The relationship between hyperuricemia and dieting remains an active research topic. Dietary purines contribute less than 20% of SUA (Basseville & Bates, 2011; Liu et al., 2019) but can be the last straw for hyperuricemia and may initiate a gout attack. Dietary food restriction is a basic requirement for hyperuricemia treatment (FitzGerald et al., 2020) and involves two aspects: dietary ingredients and amount of dietary food. Dietary ingredients refer to the food choices included in the dietary menu; for example, seafood (Lee et al., 2022) and visceral food (Jakse et al., 2019) are common high-purine foods that can increase the SUA level. Individuals with hyperuricemia should avoid consuming high-purine foods. However, optimal dieting without dietary ingredient adjustment has not been systematically investigated in uricase-deficient animals.

It has been proven that fasting elevates the level of SUA (Gao et al., 2022a; Miyake et al., 2014). As the positive relationship between obesity and hyperuricemia has been proven by epidemiologic studies (Gong et al., 2020; Zhao & Zhao, 2022), body weight restriction has become an important and basic measure to lower SUA levels. Given that fasting increases the SUA level and dieting lowers it (likely via reducing body weight), the paradox must be settled by establishing the optimal amount of dietary food that lowers the SUA level.

SUA levels were lowered by 75% dieting

Here, we confirmed once more that fasting is ineffective for lowering SUA levels in KDY rats (Fig. 2). In contrast to wild-type SD rats, KDY rats are vulnerable to fasting and water deprivation; indeed, they will die if food or water is deprived for ≥3 days. However, fasting for 2 days is sufficient to prove the opinion that fasting increases the level of SUA (Fig. 2).

To determine the optimal daily amount food for KDY rats, the normal amounts of food they ate were recorded (Fig. 3), and the average amount of food consumed was set as 100% dieting. Based on the results, different reductions in food supply were designed. Although the SUA level is relatively stable in KDY rats (Gao et al., 2022b), the levels vary between rats, therefore, the SUA level before treatment was also set as “1”. According to the relative variation in the SUA level, 75% dietary food was screened out as optimal dieting (Fig. 3). The result coincides with the theory described in traditional Chinese medicine books, which states that 70–80% of dietary food is good for health.

Considering that dietary purines contribute <20% of SUA, 75% dieting would only lower SUA levels by <5% (0.25 × 0.2). Surprisingly, on average, dieting for 2 weeks lowered the SUA level by >20% rather than 5% (Fig. 4). These results cannot be fully accounted for by discounting the amount of food, indicating the existence of another mechanism.

Likely mechanism underlying the ability of 75% dieting to lower SUA levels

As the rats that received 75% dietary food excreted less urine uric acid with lower SUA levels (Fig. 4), the likely mechanism may involve the inhibition of uric acid synthesis. Indeed, the uric acid content was significantly decreased in most organs. In particular, the uric acid content in organs, including the liver, duodenum, ileum, and kidney, which contribute large amounts of uric acid (Gao et al., 2022b), was dramatically lowered (Fig. 5). Simultaneously, Xdh, a key enzyme in uric acid synthesis in the liver, was significantly down-regulated by 75% dieting (Table 1). The kidney is the main organ for uric acid excretion; however, the expression changes of genes known to participate in uric acid transportation caused by 75% dieting could not explain the SUA lowering and the decrease in uric acid excretion.

Therefore, the decrease in the SUA level caused by 75% dieting can be mainly attributed to the decrease in uric acid synthesis, which resulted from the decrease in endogenous purines (Liu et al., 2019). Based on the results of the KEGG pathway enrichment analysis, the cell cycle (ko04110) was the top pathway down-regulated by 75% dieting in both the liver (Table 2) and kidney (Table 3). Some of the main factors, including Casp3 (participating in apoptosis; Shi, 2002), Cdk1 (factor driving the cell cycle (Chymkowitch & Enserink, 2013)), and Ki67 (promoting cell proliferation (Sun & Kaufman, 2018)), were verified at both the mRNA and protein levels in the liver (Fig. 7). This finding suggests that the slower cell cycle leads to a slower cell turnover. Given that endogenous purines are mainly generated during cell turnover (Liu et al., 2019), the decreased substrate (xanthine, which all endogenous purines can be transformed to) will result in decreased product (uric acid).

Therefore, the mechanism by which 75% dieting lowers the SUA level can be summarized as the down-regulation of cell cycle and cell turnover, resulting in less substrate for Xdh to synthesize uric acid, which ultimately lowers the SUA level (Fig. 13).

Figure 13 The underlying mechanism of 75% dieting in lowering serum uric acid (SUA) level.

Organ injuries were alleviated by 75% dieting

It has been reported that uricase deficiency causes multiple mild organ injuries in KDY rats (Fan et al., 2021), which was further verified in the present study. The possible mechanisms underlying these injuries may be associated with Uox, the protein of uricase, or the elevation of SUA, which requires further investigation. The injured organs included the kidney, liver, and intestinal tract (Fig. 8–Fig. 11), all of which can be largely alleviated by 75% dieting. Given that 75% dieting cannot repair the deficient uricase in KDY rats, the results suggest that the organ injuries are caused, at least in part, by the increase in SUA level.

Limitations and prospects

It has been proven that fasting and starvation significantly down-regulate the cell cycle pathway (Fabre et al., 2019). However, fasting for 2 days elevates rather than lowers SUA levels. The down-regulation of the cell cycle pathway may not fully explain how 75% dieting decreases the SUA level, and other factors (e.g., uric acid release from cells or tissues) affecting SUA should be unveiled soon. As it happens, 75% dieting may have less effect on uric acid release than fasting.

Additionally, the 75% dieting would have a curative effect on lowering uric acid because it reduces the urate content in multiple organs; nevertheless, its chronic effects in humans require further investigation, although notably it may be suitable to lower SUA in obese adults.

Conclusions

Our results indicate that satiety of food intake tends to elevate the level of SUA. As satiety of food intake facilitates obesity, our findings support the positive correlation between obesity and hyperuricemia. Dieting is widely accepted to be an effective measure to control obesity. However, severe dieting, especially fasting, is not good for controlling SUA levels. Taken together, the present study demonstrates that 75% dietary food lowers the SUA level and alleviates organ injuries in KDY rats, the mechanism of which is associated with the down-regulation of the cell cycle pathway.

Supplemental Information

Supplemental Information 1 ARRIVE Checklist 2.0

Click here for additional data file.

Supplemental Information 2 Raw data for Fig. 2A

Click here for additional data file.

Supplemental Information 3 Raw data for Fig. 2B

Click here for additional data file.

Supplemental Information 4 Raw data for Fig. 2C

Click here for additional data file.

Supplemental Information 5 Raw data for Fig. 2D

Click here for additional data file.

Supplemental Information 6 Raw data for Fig. 2E

Click here for additional data file.

Supplemental Information 7 Raw data for Fig. 2F

Click here for additional data file.

Supplemental Information 8 Raw data for Fig. 3A

Click here for additional data file.

Supplemental Information 9 Raw data for Fig. 3B

Click here for additional data file.

Supplemental Information 10 Raw data for Fig. 3C

Click here for additional data file.

Supplemental Information 11 Raw data for Fig. 3D

Click here for additional data file.

Supplemental Information 12 Raw data for Fig. 4A

Click here for additional data file.

Supplemental Information 13 Raw data for Fig. 4B

Click here for additional data file.

Supplemental Information 14 Raw data for Fig. 4C

Click here for additional data file.

Supplemental Information 15 Raw data for Fig. 4D

Click here for additional data file.

Supplemental Information 16 Raw data for Fig. 4E

Click here for additional data file.

Supplemental Information 17 Raw data for Fig. 4F

Click here for additional data file.

Supplemental Information 18 Raw data for Fig. 4G

Click here for additional data file.

Supplemental Information 19 Raw data for Fig. 4H

Click here for additional data file.

Supplemental Information 20 Raw data for Fig. 4I

Click here for additional data file.

Supplemental Information 21 Raw data for Fig. 4J

Click here for additional data file.

Supplemental Information 22 Raw data for Fig. 5

Click here for additional data file.

Supplemental Information 23 Raw data for Fig. 6A

Click here for additional data file.

Supplemental Information 24 Raw data for Fig. 6B

Click here for additional data file.

Supplemental Information 25 Raw data for Fig. 6C

Click here for additional data file.

Supplemental Information 26 Raw data for Fig. 6D

Click here for additional data file.

Supplemental Information 27 Raw data for Fig. 6E

Click here for additional data file.

Supplemental Information 28 Raw data for Fig. 6F

Click here for additional data file.

Supplemental Information 29 Raw data for Fig. 6G

Click here for additional data file.

Supplemental Information 30 Raw data for Fig. 7A and 7C

Click here for additional data file.

Supplemental Information 31 Raw data for Fig. 7B beta actin (first two lanes)

Click here for additional data file.

Supplemental Information 32 Raw data for Fig. 7B Casp3 (first two lanes)

Click here for additional data file.

Supplemental Information 33 Raw data for Fig. 7B CDK1 (first two lanes)

Click here for additional data file.

Supplemental Information 34 Raw data for Fig. 7B Ki67 (first two lanes)

Click here for additional data file.

Supplemental Information 35 Raw data for Table 1

Click here for additional data file.

Supplemental Information 36 Raw data for Table 2

Click here for additional data file.

Supplemental Information 37 Raw data for Table 3

Click here for additional data file.

Additional Information and Declarations

Competing Interests

Author Contributions

Animal Ethics

Data Availability

The authors declare there are no competing interests.

Yun Yu performed the experiments, authored or reviewed drafts of the article, and approved the final draft.

Xulian Wan performed the experiments, prepared figures and/or tables, authored or reviewed drafts of the article, and approved the final draft.

Dan Li performed the experiments, prepared figures and/or tables, and approved the final draft.

Yalin Qi performed the experiments, prepared figures and/or tables, and approved the final draft.

Ning Li analyzed the data, prepared figures and/or tables, and approved the final draft.

Guangyun Luo analyzed the data, authored or reviewed drafts of the article, and approved the final draft.

Hua Yin analyzed the data, prepared figures and/or tables, and approved the final draft.

Lei Wang performed the experiments, prepared figures and/or tables, and approved the final draft.

Wan Qin performed the experiments, prepared figures and/or tables, and approved the final draft.

Yongkun Li analyzed the data, prepared figures and/or tables, and approved the final draft.

Lvyu Li conceived and designed the experiments, authored or reviewed drafts of the article, and approved the final draft.

Weigang Duan conceived and designed the experiments, prepared figures and/or tables, authored or reviewed drafts of the article, and approved the final draft.

The following information was supplied relating to ethical approvals (i.e., approving body and any reference numbers):

The Animal Care and Use Committee of Kunming Medical University (Approval No. KMMU-2021578).

The following information was supplied regarding data availability:

The raw data is available in the Supplemental Files.

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
