# Peer review of "Dieting alleviates hyperuricemia and organ injuries in uricase-deficient rats via down-regulating cell cycle pathway"

_PeerJ, doi:10.7717/peerj.15999_

## Round 0.1 · original submission · Major Revisions

Dear Dr. Yu and colleagues:

Thanks for submitting your manuscript to PeerJ. I have now received two independent reviews of your work, and as you will see, the reviewers raised some concerns about the research. Despite this, these reviewers are optimistic about your work and the potential impact it will have on research studying Hyperuricemia and organ Injuries. Thus, I encourage you to revise your manuscript, accordingly, taking into account all of the concerns raised by both reviewers.

Importantly, please ensure that an English expert has edited your revised manuscript for content and clarity.

It seems your work is a continuation from a previously published paper (line 592, Qi, Y. 2021, which the reviewers were unable to access ). Unfortunately, this precluded assessment of the novelty of data from Figure 2 to Figure 4 and Figure 6. But based on the abstract of the reference, there appear to be some overlapping discoveries, such as the fasting experiment (Fig 2), measuring the body weight, water consumption, and SUA levels (Fig 3). And the TC and TG level in serum decreased after two-week treatment (Fig 6) is also published in the reference.

In your revision, please provide access to the paper for the reviewers to evaluate the work better. Please also ensure that the findings in your current work are clearly delineated from those of your previous report.

Please expand on your results more in the Discussion/Conclusion sections.

Please also note that reviewer 1 has included a marked-up version of your manuscript.

Good luck with your revision,

Best,

-joe

Reviewer 1 ·

Basic reporting

Yun et.al determined that 75% dietary intake decreased the serum uric acid (SUA) with UOX -/+ rat model and rats with 75% dietary for two weeks have been found with lower uric acid content in organs, lower triglyceride and cholesterol and organ injury alleviating.
In general, the language is clear and professional. The literature is well-referenced, and the figures are high quality and well-described.
Here are some aspects that could be improved:
1) A few minor errors such as missing spaces, extra spaces, or unnecessary commas are highlighted in the attachment.
2) Please remove all the figure and table captions from the main manuscript.
3) Line 61-65. The sentence is fragmented. It misses the be-verb but has unnecessary ‘and’. It can be written as: ‘However drugs such as Xdh inhibitor (ref), febuxostat (ref), uricosuric agents (ref), and benzbromarone (ref) to lower SUA are highly dependent on absorption, which can lead to serious [severe] adverse reactions (ref).
4) Line 141: In fasting experiments, 12 45-day old male KDY rats… --> twelve 45-day-old male..
5) Line 299: At the same time, the body weights of rats slightly but significantly decreased after 1 week of treatment… ‘slightly but significantly’ is a controversial statement.
6) Figure 5: X-axis label for the skeletal muscle is incomplete.
7) Figure 13: brocken nuclei --> broken nuclei.

Experimental design

- standard deviation: when calculating the standard deviation for a sample from a population use STDEV.S, but if calculating the standard deviation for a whole population, STDEV.P is the correct one.

Validity of the findings

- The work discussed in the manuscript is a continuing work from a previously published paper (line 592, Qi, Y. 2021). The reference is published in a local journal, please provide the pdf of the reference for novelty validation.

Annotated reviews are not available for download in order to protect the identity of reviewers who chose to remain anonymous.

Reviewer 2 ·

Basic reporting

In this study, authors found dieting is a basic treatment for lowering hyperuricemia. In this study, The optimal amount of dietary food (75% dietary intake) that lowered SUA was screen out by controlling the amount of food daily from 25% to 100% for 2 weeks. In addition to lowering SUA by approximately (22.5±20.5)%, the optimal amount of dietary food for 2 weeks inhibited urine uric acid excretion, lowered uric acid content in multiple organs, improved renal function, lowered serum triglyceride, alleviated organ injuries (including the liver, kidney and intestinal tract) at the histological level, and down-regulated the Kyoto Encyclopedia of Genes and Genome (KEGG) pathway of cell cycle (ko04110). Taken together, these results demonstrated that 75% dietary food eûectively lowers the SUA level without modifying dietary ingredients and alleviate the injuries resulting from uricase deûciency or hyperuricemia, the mechanism of which is associated with the down-regulation of cell cycle pathway.

Experimental design

This manuscript provides a reference value for lowering hyperuricemia development enough to be accepted. However, take into consideration the following points:

1. The author should take a good look at formatting, such as line 29
2. There are some minor mistakes that should have caught the writer's attention, such as line 211, replace ‘ P<0.05’ with ‘P<0.05’. Line217 , ‘25 mmol/l’ and 5,000 g, mean+/-SD and the so on. unit errors in the full text should be checked and corrected.
3. Line 47 space between 2022(Yang, Zhang, & Zhou, 2022). Make sure throughout the manuscript there is a space between the text and the reference. Please check the full references.

Validity of the findings

4. The conclusions section were written too succinctly, the authors can enrich the content.
5. The manuscript should be reviewed by a fluent English speaker.

---

## Round 0.2 · accepted · Accept

Dear Dr. Yu and colleagues:

Thanks for revising your manuscript based on the concerns raised by the reviewers. I now believe that your manuscript is suitable for publication. Congratulations! I look forward to seeing this work in print, and I anticipate it being an important resource for groups studying Hyperuricemia and organ injuries. Thanks again for choosing PeerJ to publish such important work.

Best,

-joe

Reviewer 1 ·

Basic reporting

Thanks for submitting the revision and addressing reviews' comments accordingly. There are no more concerns about the manuscript. Line 357 missed a space between treated and following and line 816-819 has a typo. Recommend accepting the manuscript after correcting the format and typo.

Experimental design

n/a

Validity of the findings

n/a

Reviewer 2 ·

Basic reporting

All revision points are correctly addressed and the paper can be considered for publishing in its present form.

Experimental design

no comment

Validity of the findings

no comment